

# Ordinal losses for classification of cervical cancer risk

Tomé Albuquerque[1,2], Ricardo Cruz[1,2] and Jaime S. Cardoso[1,2]

[1] Institute for Systems and Computer Engineering, Technology and Science, Porto, Portugal
[2] Faculty of Engineering of the University of Porto, Porto, Portugal

## ABSTRACT

Cervical cancer is the fourth leading cause of cancer-related deaths in women, especially in low to middle-income countries. Despite the outburst of recent scientific advances, there is no totally effective treatment, especially when diagnosed in an advanced stage. Screening tests, such as cytology or colposcopy, have been responsible for a substantial decrease in cervical cancer deaths. Cervical cancer automatic screening via Pap smear is a highly valuable cell imaging-based detection tool, where cells must be classified as being within one of a multitude of ordinal classes, ranging from abnormal to normal. Current approaches to ordinal inference for neural networks are found to not sufficiently take advantage of the ordinal problem or to be too uncompromising. A non-parametric ordinal loss for neuronal networks is proposed that promotes the output probabilities to follow a unimodal distribution. This is done by imposing a set of different constraints over all pairs of consecutive labels which allows for a more flexible decision boundary relative to approaches from the literature. Our proposed loss is contrasted against other methods from the literature by using a plethora of deep architectures. A first conclusion is the benefit of using non-parametric ordinal losses against parametric losses in cervical cancer risk prediction. Additionally, the proposed loss is found to be the top-performer in several cases. The best performing model scores an accuracy of 75.6% for seven classes and 81.3% for four classes.

# INTRODUCTION

The survival rate for women with cervical cancer is disturbing–in the USA, the 5-year survival rate for all women with cervical cancer is just 66% and is responsible for around 10 deaths per week in women aged 20 to 39 years (*Siegel, Miller & Jemal, 2020*). The main factor for the high mortality rate is the asymptomatic characteristic of cervical cancer in its initial stages, which justifies the need for early diagnosis. Screening tests have been responsible for a strong decrease in cervical cancer deaths. The screening programs are implemented in most developed countries and the process includes the human papillomavirus (HPV) test, the cytology test (or Pap smear), colposcopy, and biopsy (*WHO, 2019*). HPV is a group of viruses known to influence the risk of cervical cancer–some types of HPV viruses produce dysplastic changes in cells that can progressively lead to the development of cancer (*WHO, 2019*).

Corresponding author
Tomé Albuquerque,
tome.m.albuquerque@inesctec.pt

A cervical cytology test is used to detect potentially abnormal cells from the uterine cervix. These premalignant dysplastic changes of cells are classified in progressive stages: seven stages by the World Health Organization classification (WHO) system or four stages by The Bethesda classification system (TBS) (*DeMay, 2007*).

The risk of developing cancer is especially pronounced for the later stages. Therefore, distinguishing between the stages can be crucial for diagnosis. Yet, most of the literature focuses on binary classification (normal or abnormal), ignoring the fine-grained classification of cervical cells into different stages.

The classification of observations into naturally ordered classes, as the stages of the premalignant dysplastic changes, are traditionally handled by conventional methods intended to classify nominal classes where the order relation is ignored. This paper introduces a new machine learning paradigm intended for multi-class classification problems where the classes are ordered. A non-parametric loss for ordinal data classification is proposed whose goal is to promote unimodality in the prediction distributions produced by the neural network; e.g., it would be inconsistent to predict that stage 1 and stage 3 are both more likely than stage 2. Yet, this loss is more flexible than other losses from the literature that force a binomial distribution in the output (*Costa & Cardoso, 2005*). This loss is also contrasted with the standard cross-entropy loss and networks that predict classes in the form of an ordinal encoding (*Cheng, Wang & Pollastri, 2008*). The Herlev dataset, which comprises 917 images of individual cervical cells in different stages of the disease, is used in the experiments (*Jantzen & Dounias, 2006*) together with a plethora of CNN architectures.

In the next section, the problem and dataset at hand are presented. Other work for Pap smear cell classification is then reviewed in the ''Related Work'' section. The proposed loss is elaborated on the ''Proposal'' section, and the experimental details are described in ''Experiments'' with results and discussion presented in ''Results''. The study finished with a ''Conclusion'' section.

## BACKGROUND

According to the WHO classification system, there are seven different types of Pap smear cells in cervical cancer progression. This system assumes the existence of three different types of normal cells and four different types of abnormal cells. From suspicious cells to carcinoma in situ (CIS), the premalignant dysplastic changes of cells include four stages, which are mild, moderate, severe dysplasia, and carcinoma in situ (*Suhrland, 2000*). However, nowadays, the most used classification system is the TBS classification system, which is widely accepted by the medical society. According to the TBS system, the Pap smear cells can be divided into four classes: normal, Low-grade Squamous Intraepithelial Lesion (LSIL), High-grade Squamous Intraepithelial Lesion (HSIL), and Carcinoma in situ (*Nayar & Wilbur, 2017*).

The different stages of cervical cytology abnormalities are associated with different morphological changes in the cells including the cytoplasm and nucleus. However, the small visual differences between some stages of cervical cells make the construction of a multi-class autonomous classification system a real challenge.

**Table 1** The seven different Pap smear classes in the Herlev dataset.

|  | WHO | TBS | Type of cell | Quantity |
|---|---|---|---|---|
| **Normal** | 1 | 1 | Superficial squamous epithelial | 74 cells |
|  | 2 | 1 | Intermediate squamous epithelial | 70 cells |
|  | 3 | 1 | Columnar epithelial | 98 cells |
| **Abnormal** | 4 | 2 | Mild squamous non-keratinizing dysplasia | 182 cells |
|  | 5 | 3 | Moderate squamous non-keratinizing dysplasia | 146 cells |
|  | 6 | 3 | Severe squamous non-keratinizing dysplasia | 197 cells |
|  | 7 | 4 | Squamous cell carcinoma in situ intermediate | 150 cells |

**Table 2** Image examples of the seven different Pap smear classes in the Herlev dataset.

| | Normal | | | Abnormal | | | |
|---|---|---|---|---|---|---|---|
| **WHO** | $k=1$ | $k=2$ | $k=3$ | $k=4$ | $k=5$ | $k=6$ | $k=7$ |
| **TBS** | $k=1$ | | | $k=2$ | | $k=3$ | $k=4$ |

The dataset used in this work is the Herlev Dataset, which is a publicly available dataset (http://mde-lab.aegean.gr/index.php/downloads) collected at the Herlev University Hospital (Denmark) using a digital camera and microscope with an image resolution of 0.201 μm per pixel (*Jantzen & Dounias, 2006*). The preparation of the specimens followed the traditional Pap smear and Pap staining. To amplify the certainty of diagnosis, two cytotechnicians and a doctor characterized the cervical images in the Herlev dataset into seven classes. The Herlev dataset is composed of a total of 917 images of individual cervical cells. Each image contains ground truth segmentation and classification label. Table 1 shows the nomenclature of the seven different classes from the dataset, wherein classes 1–3 correspond to types of normal cells and classes 4–7 to different levels of abnormal cells. Illustrations of these classes are then displayed in Table 2.

In most cases, the abnormal cells present a nucleus size bigger than healthy cells. However, the difference between the normal columnar nucleus and severe and/or carcinoma nucleus is not easy to differen tiate, which makes the classification between these different types of cells a challenge.

There is some imbalance in the class distribution of the dataset: 8%, 7%, 11%, 19%, 16%, 22%, and 17%, whereas 14% would be expected if the distribution was uniform.

## RELATED WORK

In most literature, the classification of Pap smear images consists of a binary separation between normal and abnormal cell (two classes), using different methodologies such as Support Vector Machines (SVM) (*Chen et al., 2014*; *Chankong, Theera-Umpon & Auephanwiriyakul, 2014*; *Kashyap et al., 2016*; *Bora et al., 2017*), $k$-Nearest Neighbours

(kNN) (*Chankong, Theera-Umpon & Auephanwiriyakul, 2014*; *Bora et al., 2017*; *Marinakis, Dounias & Jantzen, 2009*; *Fekri Ershad, 2019*), Fuzzy $c$-Means Algorithm (FCM) (*Chankong, Theera-Umpon & Auephanwiriyakul, 2014*; *William et al., 2019*), $k$-Means clustering (*Paul, Bhowmik & Bhattacharjee, 2015*), Artificial Neural Networks (ANN) (*Chankong, Theera-Umpon & Auephanwiriyakul, 2014*), and, more recently, Convolutional Neural Networks (CNN) (*Zhang et al., 2017*; *Lin et al., 2019*; *Kurnianingsih et al., 2019*).

However, all this work consists of binary classification, which is useful for screening, but not enough for a confident diagnosis. Fewer works explore the multi-class classification of cervical cells on the Herlev dataset (*Chankong, Theera-Umpon & Auephanwiriyakul, 2014*). proposed a multi-class automatic cervical cancer cell classification system using different classifiers, such as FCM, ANN, and kNN. However, this system is based only on 9 cell-based features. The approach applies feature extraction from the nucleus and cytoplasm in each image and requires manual selection of the best threshold to minimize the error when applying the classifier to construct the cell mask. More recently (*Kurnianingsih et al., 2019*), perform feature extraction in a more autonomous way using a CNN. The use of a CNN simplifies the pre-processing steps that were necessary for the approach by Chankong et al. *Ghoneim, Muhammad & Hossain (2019)* proposed a new approach for multi-class cervical cancer cell detection and classification, using in the first step, CNNs to extract deep-learned features and in the second step, extreme learning machine (ELM)-based classifiers to classify the input cell images. *Lin et al. (2019)* proposed a new CNN-based method that combines cell image appearance with cell morphology for multi-class classification of cervical cells in the Herlev dataset. In all these cases, cross-entropy is adopted for ordinal data classification.

Assume that examples in a classification problem come from one of K classes, labelled from $\mathcal{C}^{(1)}$ to $\mathcal{C}^{(K)}$, corresponding to their natural order in ordinal classes, and arbitrarily for nominal classes.

**Cross-Entropy (CE):** Traditionally, a CNN would perform multi-class classification by minimizing cross-entropy, averaged over the training set,

$$\text{CE}(\mathbf{y}_n, \hat{\mathbf{y}}_n) = -\sum_{k=1}^{K} y_{nk} \log(\hat{y}_{nk}),$$

where $\mathbf{y}_n = [y_{n1} \cdots y_{nk} \cdots y_{nK}] \in R^K$ represents the one-hot encoding of the class of the $n$-th observation and $\hat{\mathbf{y}}_n = [\hat{y}_{n1} \cdots \hat{y}_{nk} \cdots \hat{y}_{nK}] \in R^K$ is the output probability vector given by the neural network for observation $n$. Note that $y_{nk} \in \{0, 1\}$, $\hat{y}_{nk} \in [0, 1]$ and $\sum_{k=1}^{K} y_{nk} = \sum_{k=1}^{K} \hat{y}_{nk} = 1$.

However, CE has limitations when applied to ordinal data. Defining $k_n^\star \in \{1, \cdots, K\}$ as the index of the true class of observation $\mathbf{x}_n$ (the position where $y_{nk} = 1$), it is then clear that

$$\text{CE}(\mathbf{y}_n, \hat{\mathbf{y}}_n) = -\log(\hat{y}_{nk_n^\star}).$$

Intuitively, CE is just trying to maximize the probability in the output corresponding to the true class, ignoring all the other probabilities. For this loss, an error between classes $\mathcal{C}^{(1)}$ and $\mathcal{C}^{(2)}$ is treated as the same as an error between $\mathcal{C}^{(1)}$ and $\mathcal{C}^{(K)}$, which is undesirable for ordinal problems.

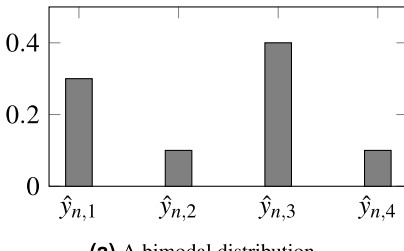

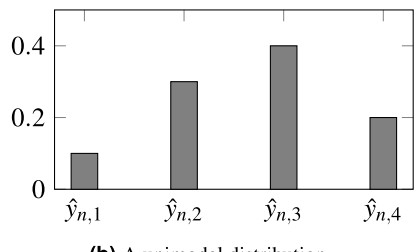

| (a) A bimodal distribution | (b) A unimodal distribution |

**Figure 1** **Probabilities produced by two different models for observation *n*.** (A) Biomodal distribution. (B) Unimodal distribution. CE is unable to distinguish both scenarios, setting the same loss for both. For ordinal problems, a unimodal distribution, peaking in the true class, is, arguably, preferable. In this example, $k_n^\star = 3$ is the assumed true class.

Furthermore, the loss does not constrain the model to produce unimodal probabilities, so inconsistencies can be produced such as $\hat{y}_{nj} > \hat{y}_{n\ell} < \hat{y}_{ni}$, even when $1 \leq j < \ell < i \leq K$. It would be preferable for output probabilities to follow a unimodal distribution, as depicted by Fig. 1.

Cross-entropy is a fair approach for nominal data, where no additional information is available. However, for ordinal data, the order can be explored to further regularize learning.

**Ordinal encoding (OE):** A model agnostic way to introduce ordinality is by training binary classifiers, in the form of an ensemble, where each classifier tries to distinguish between each pair of adjacent classes, $\mathcal{C}^{(i)}$ and $\mathcal{C}^{(i+1)}$ (*Frank & Hall, 2001*). An adaptation for neural networks consists of training a single neural network to produce $K-1$ outputs, where each output makes a binary decision between each pair of adjacent classes. The information on the ordinal distribution can, therefore, be encoded in the **y** labels themselves (*Cheng, Wang & Pollastri, 2008*).

In traditional one-hot encoding, classes are encoded using the indicator function $\mathbb{1}(k = k^\star)$, so that $y_{nm}$ is represented by 1 if $k = k_n^\star$ and 0 otherwise. In ordinal encoding, classes are encoded using a cumulative distribution –the indicator function used is $\mathbb{1}(k < k^\star)$ so that $y_{nm}$ is represented by 1 if $k < k_n^\star$ and 0 otherwise. Each output represents the incremental neighbor probability, and the inverse operation (during inference) is performed by summing up these outputs, $p_{nk} = \sum_{m=1}^{K-1} y_{nm}$.

**Unimodal (U):** Another method to promote ordinality in classification problems consists of constraining discrete ordinal probability distributions to be unimodal using binomial or Poisson probability distributions:

–> **Binomial Unimodal (BU):** One approach is to constrain the output of the network directly, approaching the problem under a regression setting. Instead of several outputs, the output predicts a single output representing the probability along the classes, with $y_n = 0$ representing $k_n^\star = 1$ and $y_n = 1$ representing $k_n^\star = K$ (*Costa & Cardoso, 2005*; *Beckham & Pal, 2017*). Thus, this model has only one output unit as the final layer. The model's sigmoid output is converted into class probabilities using the Binomial

probability mass function. The goal of this approach is to maintain the ordinality of the classes by applying a parametric model for the output probabilities.

–> **Poisson Unimodal (PU):** The Poisson probability mass function (PMF) is used to enforce a discrete unimodal probability distribution (*Beckham & Pal, 2017*). As a final layer, the log Poisson PMF transform is applied together with a softmax to normalize the output as a probability distribution.

The major difference between *Costa & Cardoso (2005)* and *Beckham & Pal (2017)* is that explore Binomial/Poisson distributions in the context of deep learning (rather than classical machine learning approaches), *Beckham & Pal (2017)* also propose the use of a learnable softmax temperature term to control the variance of the distribution. In the experiments, the temperature term ($\tau$) was used as a constant value of 1.

These parametric approaches sometimes sacrifice accuracy to ensure the ordinality assumption. This sacrifice might sometimes prove too much, especially given the fact that modern deep learning datasets are massive and have a significant number of mislabeled examples. A loss is now proposed to stimulate a unimodal output without modifying the network architecture.

## PROPOSAL

As already explored, CE presents drawbacks when applied to ordinal data. By focusing only on the mode of the distribution and ignoring all the other values in the output probability vector, one is not leveraging the ordinal information intrinsic to the data.

### Fixing CE with an ordinal loss term

A possible fix for CE is to add a regularization term that penalizes the deviations from the unimodal setting. Defining $\mathbb{1}(x)$ as the indicator function of $x$ and $\text{ReLU}(x) = x\mathbb{1}(x > 0) = \max(0, x)$, a tentative solution for an order-aware loss could be

$$\text{CO}(\mathbf{y}_n, \hat{\mathbf{y}}_n) = \text{CE}(\mathbf{y}_n, \hat{\mathbf{y}}_n) + \lambda \sum_{k=1}^{K-1} \mathbb{1}(k \geq k_n^\star) \text{ReLU}(\hat{y}_{n(k+1)} - \hat{y}_{n(k)})$$

$$+ \lambda \sum_{k=1}^{K-1} \mathbb{1}(k \leq k_n^\star) \text{ReLU}(\hat{y}_{n(k)} - \hat{y}_{n(k+1)}), \quad (1)$$

where $\lambda \geq 0$ controls the relative importance of the extra terms favoring unimodal distributions. Predicted probability values are expected to decrease monotonously as we depart left and right from the true class. The added terms penalize any deviation from this expected unimodal distribution, with a penalty proportional to the difference of the consecutive probabilities. The additional terms, although promoting uni-modality, still allow flat distributions. A generalization of the previous idea is to add a margin of $\delta > 0$ to the ReLU, imposing that the difference between consecutive probabilities is at least $\delta$. This

leads us to a second CE loss, CO2, suitable for ordinal classes:

$$\text{CO2}(\mathbf{y}_n, \hat{\mathbf{y}}_n) = \text{CE}(\mathbf{y}_n, \hat{\mathbf{y}}_n) + \lambda \sum_{k=1}^{K-1} \mathbb{1}(k \geq k_n^{\star}) \text{ReLU}(\delta + \hat{y}_{n(k+1)} - \hat{y}_{n(k)})$$

$$+ \lambda \sum_{k=1}^{K-1} \mathbb{1}(k \leq k_n^{\star}) \text{ReLU}(\delta + \hat{y}_{n(k)} - \hat{y}_{n(k+1)}). \quad (2)$$

A value of $\delta = 0.05$ has been empirically found to provide a sensible margin. This loss is aligned with the proposal present in *Belharbi et al. (2019)*.

**Beyond CO2: ordinal entropy loss function**

In CO2, the CE term by itself is only trying to maximize the probability estimated in the true output class (while ignoring the remaining probabilities); the ordinal terms are promoting unimodality but not penalizing (almost) flat distributions. This also explains why the ordinal terms by themselves (especially the version without margin) are not enough to promote strong learning: the model could converge to solutions where the predicted probability in the true class is only slightly above the neighbouring probabilities, which will not, most likely, provide a strong generalization for new observations.

However, the extreme nature of CE, ignoring almost everything in the predicted distribution $\hat{\mathbf{y}}_n$ is equivalent to assuming that the perfect probability distribution is one on the true class and zero everywhere else. This assumes a strong belief and dependence on the chosen one-hot encoding, which is often a crude approximation to the true probability class distribution. Seldom, for a fixed observation $\mathbf{x}_n$, the class is deterministically known; rather, we expect a class distribution with a few non-zero values. This is particularly true for observations close to the boundaries between classes. A softer assumption is that the distribution should have a low entropy, only.

This leads us to propose the ordinal entropy loss, HO2, for ordinal data as

$$\text{HO2}(\mathbf{y}_n, \hat{\mathbf{y}}_n) = \text{H}(\hat{\mathbf{y}}_n) + \lambda \sum_{k=1}^{K-1} \mathbb{1}(k \geq k_n^{\star}) \text{ReLU}(\delta + \hat{y}_{n(k+1)} - \hat{y}_{n(k)})$$

$$+ \lambda \sum_{k=1}^{K-1} \mathbb{1}(k \leq k_n^{\star}) \text{ReLU}(\delta + \hat{y}_{n(k)} - \hat{y}_{n(k+1)}), \quad (3)$$

where $\text{H}(\mathbf{p})$ denotes the entropy of the distribution $\mathbf{p}$.

## EXPERIMENTS

Several neural network architectures are now trained using the aforementioned losses for the dataset at hand. In this work, it was also evaluated the performance differences between parametric and non-parametric losses for ordinal classification (Fig. 2). All the experiments are implemented in PyTorch and are available online (https://github.com/tomealbuquerque/ordinal-losses).

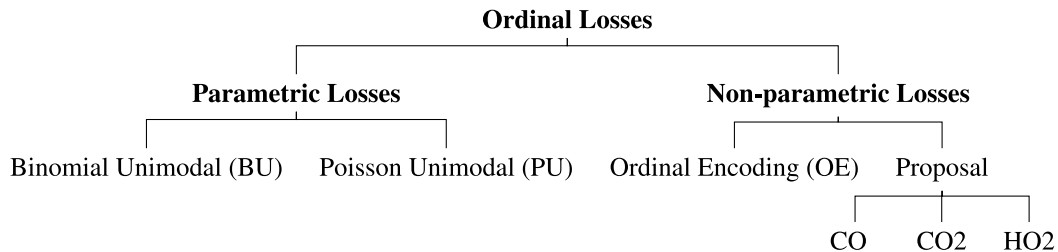

**Figure 2  Schematic representation of the used and proposed ordinal losses.**

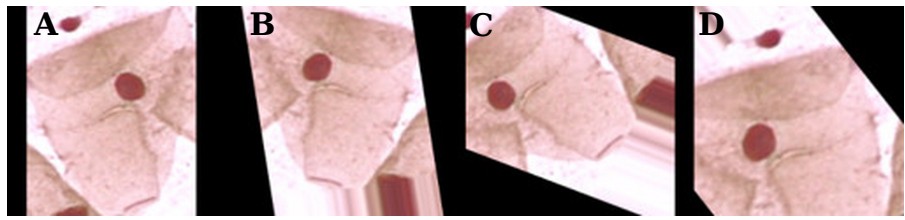

**Figure 3  Examples of data augmentation on the Herlev database.** The original zero-padding image (A) and random transformations (B–D).

## Data pre-processing

Given that all images from the Herlev dataset are of different sizes, all images were resized to $224 \times 224$ pixels; however, before the resize of cytological images, a zero-padding must be done to avoid the loss of essential information regarding cells shape. The last pre-processing step was to apply the same normalization as used by ImageNet (*Simonyan & Zisserman, 2014*).

Since the Herlev database has a relatively small number of observations (917), the training dataset was augmented by a series of random transformations: 10% of width and height shift, 10% of zoom, image rotation, horizontal and vertical flips, and color saturation. These transformations are illustrated in Fig. 3.

## Convolutional neural networks

A convolutional neural network (CNN) is a neural network that successively applies convolutions of filters to the image. These filters are learned and consist of quadrilateral patches that are convolved across the whole input image—unlike previous fully-connected networks, only local inputs are connected at each layer. Typically, each convolution is intertwined with downsampling operations, such as max-pooling, that successively reduce the size of the original image.

The final layers are fully-connected and then the final output is processed by a soft-max for multi-class problems or a logistic function for binary classification. Dropout was used to reduce overfitting by constraining these fully-connected layers (*Srivastava et al., 2014*).

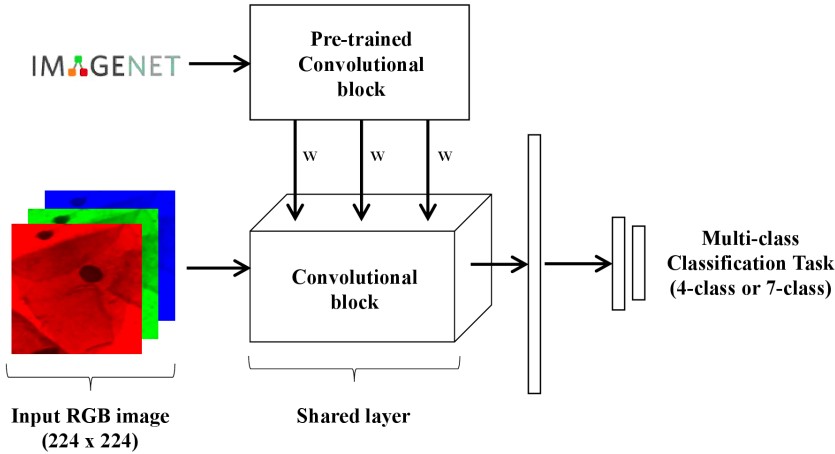

**Figure 4** Schematic representation of the model used for multi-class classification of Pap smear cells.

## Network architectures

Two different models were trained and tested in this work for multi-class (4-class and 7-class) classification of Pap smear cells images (Fig. 4). Both models were trained and tested with nine different convolutional network architectures: AlexNet (*Krizhevsky, Sutskever & Hinton, 2012*), GoogLeNet (*Szegedy et al., 2015*), MobileNet_V2 (*Howard et al., 2017*), ResNet18 (*He et al., 2016*), ResNeXt50_32X4D (*Xie et al., 2017*), ShuffleNet_V2_X1_0 (*Zhang et al., 2017*), SqueezeNet1_0 (*Iandola et al., 2016*), VGG-16 (*Simonyan & Zisserman, 2014*), and Wide_ResNet50_2 (*Zagoruyko & Komodakis, 2016*). The goal of testing these different architectures is to evaluate how well the proposed loss behaves in a wide range of architectures. These nine different architectures were chosen as they are often used in the literature and came pre-trained with PyTorch on ImageNet (https://pytorch.org/docs/stable/torchvision/models.html). The last block of each architecture was replaced by the following layers: dropout with $p = 20\%$, 512-unit dense layer with ReLU, dropout with $p = 20\%$, a 256-wide dense layer with ReLU, followed by $K$ output neurons.

A brief introduction of each architecture is now presented. AlexNet, based on LeNet, formalized the Convolutional Neural Network (CNN) as is known today: a series of convolutions intertwined by downsampling blocks. Max-pooling was used for downsampling and ReLU was used as the activation function. It became famous for winning ImageNet, the first CNN to do so (*Krizhevsky, Sutskever & Hinton, 2012*). The following ImageNet competitions were also won by other CNNs–VGG and GoogLeNet–which were evolutions on top of AlexNet that consist mostly of a much higher number of parameters (*Simonyan & Zisserman, 2014*; *Szegedy et al., 2015*). Then, MobileNet (*Howard et al., 2017*) introduced hyperparameters to help the user choose between latency and accuracy trade-offs. An attempt was then made at curbing the number of parameters with ShuffleNet (*Zhang et al., 2018*) by approximating convolution operators using fewer parameters.

Finally, an attempt was made at curbing the number of parameters, which had been exploding, while keeping the accuracy of these early CNNs with SqueezeNet (*Iandola et al., 2016*).

In another line of research, ResNet (*He et al., 2016*) introduced residual blocks whose goal was to make the optimization process easier for gradient descent. Each residual block learns $a = f(x) + x$ instead of $a = f(x)$. Given that weights are initialized randomly around zero and most activation functions are also centred in zero (an exception would be the logistic activation function), then, in expectation, all neurons output zero before any training. Therefore, when using residual blocks, at time = 0, $a = x$, i.e., activations produce the identity function. This greatly helps gradient descent focus on finding improvements (residuals) on top of the identity function. While this model allowed for deeper neural networks, each per cent of improved accuracy required nearly doubling the number of layers, which motivated WideResNet (*Zagoruyko & Komodakis, 2016*) and ResNeXt (*Xie et al., 2017*) to improve the residual architecture to improve learning time.

### Training

The weights of the architectures previously mentioned are already initialized by pre-training on ImageNet. Adam was used as the optimizer and started with a learning rate of $10^{-4}$. The learning rate is reduced by 10% whenever the loss is stagnant for 10 epochs. The training process is completed after 100 epochs.

The dataset was divided into 10 different folds using stratified cross-validation, in order to maintain the class ratios. Therefore, the results are the average and deviation of these 10 folds. In the case of the proposed loss, the hyperparameter $\lambda$ is tuned by doing nested $k$-fold cross-validating using the training set (with $k = 5$) in order to create an unbiased validation set.

### Evaluation metrics

The most popular classification metric is accuracy (Acc). For $N$ observations, taking $k_i$ and $\hat{k}_i$ to be the label and prediction of the $n$-th observation, respectively, then $\text{Acc} = \frac{1}{N} \sum_{n=1}^{N} \mathbb{1}(\hat{k}_n^\star = k_n^\star)$, where $\mathbb{1}$ is the indicator function.

However, this metric treats all class errors as the same, whether the error is between adjacent classes or between classes in the extreme. If we have $K$ classes represented by a set $\mathcal{C} = \{\mathcal{C}^{(1)}, \mathcal{C}^{(2)}, \ldots, \mathcal{C}^{(K)}\}$, then accuracy will treat an error between $\mathcal{C}^{(1)}$ and $\mathcal{C}^{(2)}$ with the same magnitude as an error between $\mathcal{C}^{(1)}$ and $\mathcal{C}^{(K)}$ which is clearly worse. As an illustration, in a medical setting, a misdiagnosis between Stage II and Stage III of a disease, while bad, is not as bad as a misdiagnosis between Healthy and Stage III. For that reason, a popular metric for ordinal classification is the Mean Absolute Error (MAE), $\text{MAE} = \frac{1}{N} \sum_i |k_i^\star - \hat{k}_i^\star|$. This metric is not perfect since it treats an ordinal variable as a cardinal variable. An error between classes $\mathcal{C}^{(1)}$ and $\mathcal{C}^{(3)}$ will be treated as two times worse than an error between classes $\mathcal{C}^{(1)}$ and $\mathcal{C}^{(2)}$. Naturally, the assumption of cardinality is not always warranted.

To evaluate the models' performance, we also used a metric specific for ordinal classification, Uniform Ordinal Classification Index (UOC) which considers accuracy and ranking in the performance assessment and is also robust against imbalanced classes (*Silva, Pinto & Cardoso, 2018*). The better the performance, the lower the UOC.

By combining a quality assessment (accuracy) with a quantity assessment (MAE) and also with a specific metric for ordinality (UOC) we hope to provide a balanced view of the performance of the methods.

The two other metrics used are the AUC of ROC or AUROC (Area Under the Receiver Operating Characteristic) and Kendall's $\tau$ rank correlation coefficient. AUROC measures how well-calibrated are the probabilities produced by the model. This first metric is used in the binary classification context (two classes) and is extended for multi-class by comparing each class against the rest (one vs rest strategy) and performing an overall average, known as macro averaging. On the other hand, Kendall's Tau is a non-parametric evaluation of relationships between columns of ranked data, so it is a measure of ordinal association between data. The $\tau$ correlation coefficient returns a value that ranges from $-1$ to $1$, with $0$ being no correlation and $1$ perfect correlation.

## RESULTS

The average performance for the 10-folds of nine different architectures are presented in Tables 3–8 and A1 and A2, for both the 7-class and 4-class classification problems, with the seven different learning losses –conventional Cross-Entropy (CE), Binomial Unimodal (BU) (*Costa & Cardoso, 2005*), Poisson Unimodal (PU) (*Beckham & Pal, 2017*), Ordinal Encoding (OE) (*Cheng, Wang & Pollastri, 2008*) and our proposed losses (CO, CO2 and HO2), as measured by MAE, accuracy, UOC index and Kendall's coefficient detailed in the previous section. The best models are shown in bold, while italic is used to check for statistical similarity between the other models and the best one. A *p*-value of 0.1 is used with a two-sided paired *t*-test due to the small sample size (10 folds).

For the 7-class classification problem, Table 3 shows the results for MAE, which confirm the influence of ordinal losses in promoting ordinality when comparing to nominal loss (CE). OE loss achieved the best performance across the different architectures but it is also notable the good performance of our loss: in 67% of cases, the models trained with our proposed loss provide better MAE results. The MAE results present in Table 3 for 7-class classification are consistent with the 4-class Table 6, with ordinal losses winning over nominal CE.

Tables 4 and 7 present the accuracy results for 7-class and 4-class classification problems, respectively. Regarding this metric, the results between nominal and ordinal losses are more balanced. CE loss performance is above ordinal losses in 11% for the 7-class problem and is tied for the 4-class problem. This can be explained by the lower role of ordinality in the CE loss, as also confirmed by the MAE results. This means that when misclassification occurs, ordinal losses tend to classify Pap smear images as being closer to the real class. Results for UOC index (Tables 5 and 8) are also consistent with the MAE metric, with 78% of the models presenting a lowest UOC index when using the ordinal losses. Tables A1 and A2 in the appendix present the results for Kendall's $\tau$ coefficient test in 4-class and 7-class classification problems. These results are also aligned with the results of MAE and UOC metrics: the ordinal losses perform better advantage when comparing with nominal CE.

**Table 3  Results in terms of Mean Absolute Error (MAE) for seven class problem, averaged for 10 folds (lower is better).**

|  | CE | BU | PU | OE | CO | CO2 | HO2 |
|---|---|---|---|---|---|---|---|
| AlexNet | 0.46 ± 0.08 | 0.52 ± 0.09 | 0.50 ± 0.09 | *0.44 ± 0.08* | 0.90 ± 0.19 | **0.41 ± 0.08** | 0.45 ± 0.10 |
| GoogLeNet | *0.39 ± 0.05* | 0.41 ± 0.07 | 0.42 ± 0.08 | 0.38 ± 0.09 | 0.53 ± 0.10 | *0.37 ± 0.07* | **0.36 ± 0.06** |
| MobileNet_v2 | 0.34 ± 0.05 | 0.36 ± 0.04 | **0.31 ± 0.04** | 0.33 ± 0.05 | 0.52 ± 0.26 | 0.34 ± 0.06 | *0.34 ± 0.05* |
| ResNet18 | *0.34 ± 0.09* | *0.36 ± 0.06* | *0.35 ± 0.06* | 0.35 ± 0.10 | 0.49 ± 0.11 | **0.34 ± 0.07** | *0.35 ± 0.10* |
| ResNeXt50_32x4d | 0.34 ± 0.07 | *0.33 ± 0.05* | *0.33 ± 0.03* | 0.34 ± 0.06 | 0.41 ± 0.08 | *0.33 ± 0.06* | **0.31 ± 0.07** |
| ShuffleNet_v2_x1_0 | 0.41 ± 0.07 | 0.49 ± 0.07 | *0.41 ± 0.05* | *0.38 ± 0.07* | 0.47 ± 0.08 | *0.40 ± 0.05* | **0.38 ± 0.06** |
| SqueezeNet1_0 | **0.38 ± 0.07** | 0.45 ± 0.05 | 0.46 ± 0.07 | *0.40 ± 0.09* | 0.97 ± 0.31 | 0.41 ± 0.08 | 0.45 ± 0.09 |
| VGG16 | *0.37 ± 0.09* | 0.44 ± 0.05 | 0.44 ± 0.10 | *0.37 ± 0.06* | 0.67 ± 0.15 | *0.36 ± 0.06* | **0.36 ± 0.07** |
| Wide_ResNet50_2 | 0.33 ± 0.06 | 0.37 ± 0.05 | 0.32 ± 0.06 | **0.30 ± 0.04** | 0.45 ± 0.13 | 0.33 ± 0.06 | 0.35 ± 0.09 |
| **Avg** | 0.37 | 0.41 | 0.39 | 0.36 | 0.60 | 0.37 | 0.37 |
| **Winners** | 1 | 0 | 1 | 1 | 0 | 2 | 4 |

Notes.
  **Bold:** best model, *italic:* statistically similar to best (paired *t*-test).

**Table 4  Results in terms of accuracy for seven class problem, averaged for 10 folds (higher is better).**

|  | CE | BU | PU | OE | CO | CO2 | HO2 |
|---|---|---|---|---|---|---|---|
| AlexNet | **71.1 ± 5.1** | 60.6 ± 3.7 | 64.8 ± 5.4 | *70.1 ± 5.1* | 44.2 ± 7.6 | *70.8 ± 5.1* | 67.9 ± 5.4 |
| GoogLeNet | **72.5 ± 3.7** | 66.1 ± 4.3 | 68.5 ± 4.5 | 71.5 ± 5.3 | 59.7 ± 8.2 | *72.4 ± 4.9* | *72.4 ± 3.7* |
| MobileNet_v2 | **75.0 ± 4.4** | 69.0 ± 3.5 | 74.2 ± 2.8 | 74.4 ± 3.8 | 64.4 ± 16.5 | 73.1 ± 3.7 | 74.1 ± 3.9 |
| ResNet18 | **74.4 ± 6.1** | 69.5 ± 3.7 | *73.3 ± 4.3* | 73.6 ± 6.4 | 64.6 ± 6.5 | *73.3 ± 4.5* | 73.3 ± 6.4 |
| ResNeXt50_32x4d | *74.4 ± 3.7* | 72.4 ± 4.3 | 72.8 ± 2.8 | *74.0 ± 4.2* | 68.0 ± 5.9 | *75.5 ± 3.5* | **75.7 ± 5.3** |
| ShuffleNet_v2_x1_0 | **71.9 ± 5.5** | 61.0 ± 4.5 | 67.7 ± 4.6 | 70.7 ± 4.9 | 65.5 ± 4.5 | *70.7 ± 3.1* | 71.3 ± 3.7 |
| SqueezeNet1_0 | **73.0 ± 4.3** | 63.3 ± 2.4 | 67.3 ± 3.6 | *71.8 ± 5.3* | 40.5 ± 13.3 | 70.8 ± 4.5 | 67.1 ± 5.0 |
| VGG16 | **73.1 ± 4.7** | 63.9 ± 4.6 | 67.6 ± 6.2 | *72.6 ± 3.8* | 54.4 ± 8.5 | *71.8 ± 3.3* | 72.0 ± 3.7 |
| Wide_ResNet50_2 | *75.7 ± 3.2* | 69.7 ± 3.1 | 74.5 ± 4.3 | **76.8 ± 1.9** | 66.1 ± 7.8 | *75.6 ± 4.0* | 74.3 ± 5.7 |
| **Avg** | 73.4 | 66.2 | 70.1 | 72.8 | 58.6 | 72.6 | 72.0 |
| **Winners** | 7 | 0 | 0 | 1 | 0 | 0 | 1 |

Notes.
  **Bold:** best model, *italic:* statistically similar to best (paired *t*-test).

Adding the margin (CO → CO2) influences positively most of the metrics for 7 and 4 classes. Using entropy (CO2 or HO2), instead of cross-entropy, promotes better results on the metrics intrinsically connected with ordinality (MAE, UOC and Kendall's $\tau$ coefficient).

The average results for all losses across the nine different architectures for MAE, accuracy, UOC, AUROC, Kendall's $\tau$ coefficient and Gini index metrics are present Tables A3 and A4 in the appendix for 7 and 4 class classification respectively. Both tables present the results using the classical mode (softmax) to aggregate the probabilities and also using mean (expectation trick) (*Beckham & Pal, 2017*).

On average, in most metrics, non-parametric losses outperformed parametric losses. This difference can be justified with the greater flexibility in boundary decisions provided by non-parametric losses. OE, CO2 and HO2 provided better results across the different metrics when comparing to BU and PU.

**Table 5** Results in terms of the Uniform Ordinal Classification Index (UOC) for seven class problem, averaged for 10 folds (lower is better).

| | CE | BU | PU | OE | CO | CO2 | HO2 |
|---|---|---|---|---|---|---|---|
| AlexNet | *45.1 ± 6.5* | 51.7 ± 5.7 | 49.8 ± 6.6 | 44.0 ± 6.9 | 70.3 ± 7.8 | **42.8 ± 7.3** | 46.4 ± 7.8 |
| GoogLeNet | *38.9 ± 6.0* | 44.2 ± 5.7 | 44.6 ± 7.3 | 39.0 ± 7.2 | 51.3 ± 9.1 | *38.8 ± 6.9* | **38.1 ± 4.7** |
| MobileNet_v2 | 36.0 ± 5.7 | 39.7 ± 4.9 | **33.6 ± 4.5** | 35.4 ± 5.6 | 46.7 ± 15.0 | 36.2 ± 6.4 | 36.2 ± 6.1 |
| ResNet18 | **36.2 ± 9.3** | 40.1 ± 5.7 | *37.2 ± 6.3* | 37.3 ± 9.1 | 46.9 ± 6.8 | *37.1 ± 7.6* | 37.8 ± 8.7 |
| ResNeXt50_32x4d | 36.9 ± 6.8 | *37.0 ± 5.2* | 37.6 ± 4.6 | 36.8 ± 6.1 | 42.2 ± 6.7 | *35.3 ± 6.7* | **34.0 ± 7.2** |
| ShuffleNet_v2_x1_0 | 41.8 ± 7.1 | 49.6 ± 6.4 | 43.6 ± 4.9 | **40.3 ± 6.3** | 46.3 ± 6.0 | 42.4 ± 4.1 | *40.3 ± 4.9* |
| SqueezeNet1_0 | **40.4 ± 6.0** | 47.9 ± 3.8 | 47.5 ± 4.8 | *42.4 ± 8.1* | 73.6 ± 13.6 | 42.7 ± 7.4 | 46.8 ± 7.0 |
| VGG16 | **38.5 ± 8.2** | 47.2 ± 4.9 | 45.5 ± 8.6 | 39.0 ± 6.4 | 60.3 ± 10.0 | *40.2 ± 6.1* | *39.6 ± 6.8* |
| Wide_ResNet50_2 | 35.7 ± 5.2 | 40.8 ± 5.4 | 35.6 ± 6.3 | **33.5 ± 4.5** | 44.2 ± 9.1 | *34.8 ± 6.5* | 36.6 ± 8.4 |
| **Avg** | 38.8 | 44.2 | 41.7 | 38.6 | 53.5 | 39.0 | 39.5 |
| **Winners** | 3 | 0 | 1 | 2 | 0 | 1 | 2 |

Notes.
**Bold:** best model, *italic:* statistically similar to best (paired *t*-test).

**Table 6** Results in terms of the Uniform Ordinal Classification Index (UOC) for four class problem, averaged for 10 folds (lower is better).

| | CE | BU | PU | OE | CO | CO2 | HO2 |
|---|---|---|---|---|---|---|---|
| AlexNet | *0.31 ± 0.06* | 0.32 ± 0.04 | **0.28 ± 0.04** | *0.29 ± 0.06* | 0.47 ± 0.19 | *0.29 ± 0.05* | *0.31 ± 0.06* |
| GoogLeNet | *0.24 ± 0.04* | 0.25 ± 0.03 | 0.25 ± 0.05 | *0.24 ± 0.05* | 0.38 ± 0.17 | **0.22 ± 0.05** | 0.25 ± 0.06 |
| MobileNet_v2 | *0.22 ± 0.06* | **0.21 ± 0.03** | 0.24 ± 0.05 | *0.22 ± 0.06* | *0.23 ± 0.04* | 0.24 ± 0.05 | *0.22 ± 0.05* |
| ResNet18 | 0.24 ± 0.03 | 0.26 ± 0.05 | *0.24 ± 0.05* | **0.22 ± 0.04** | 0.29 ± 0.11 | *0.22 ± 0.04* | 0.26 ± 0.06 |
| ResNeXt50_32x4d | *0.21 ± 0.03* | 0.22 ± 0.04 | 0.23 ± 0.03 | **0.20 ± 0.04** | 0.28 ± 0.07 | *0.21 ± 0.03* | 0.22 ± 0.05 |
| ShuffleNet_v2_x1_0 | *0.28 ± 0.05* | 0.33 ± 0.05 | **0.27 ± 0.05** | 0.31 ± 0.06 | 0.36 ± 0.09 | *0.28 ± 0.06* | *0.28 ± 0.04* |
| SqueezeNet1_0 | *0.28 ± 0.06* | 0.30 ± 0.05 | 0.30 ± 0.06 | **0.27 ± 0.07** | 0.66 ± 0.17 | *0.29 ± 0.04* | 0.31 ± 0.05 |
| VGG16 | 0.27 ± 0.06 | 0.28 ± 0.06 | *0.26 ± 0.05* | **0.24 ± 0.03** | 0.53 ± 0.18 | 0.26 ± 0.05 | 0.27 ± 0.05 |
| Wide_ResNet50_2 | 0.23 ± 0.05 | 0.22 ± 0.04 | **0.20 ± 0.06** | *0.22 ± 0.05* | 0.43 ± 0.22 | *0.21 ± 0.05* | *0.22 ± 0.03* |
| **Avg** | 0.25 | 0.27 | 0.25 | 0.24 | 0.40 | 0.25 | 0.26 |
| **Winners** | 0 | 1 | 3 | 4 | 0 | 1 | 0 |

Notes.
**Bold**, best model, *italic:* statistically similar to best (paired *t*-test).

Most work from the literature concerns the binary case using the Herlev dataset (normal vs abnormal); only a couple concern themselves with the 7-class and 4-class ordinal classification problem. Table 9 contrasts the best performing models from two recent works against the proposed method. In our case, the non-parametric loss (CO2) was able to beat the state-of-the-art nominal-class approaches by 11.1% (7 classes) and by 10% (4 classes) in the accuracy metric. Furthermore, the confusion matrices in Fig. 5 contrast the proposal against (*Lin et al., 2019*).

There are classes of cells easier to classify than others, as shown by the confusion matrix in Fig. 5B. Columnar cells are sometimes inappropriately classified as severe dysplasia cells since severe dysplasia cells have similar characteristics in appearance and morphology with columnar cells (e.g., small cytoplasm, dark nuclei).

The main challenge occurs in the classification of abnormal cells (i.e., mild dysplasia, moderate dysplasia, severe dysplasia, and carcinoma) where the characteristics of these

**Table 7  Results in terms of accuracy for four class problem, averaged for 10 folds (higher is better).**

|  | CE | BU | PU | OE | CO | CO2 | HO2 |
|---|---|---|---|---|---|---|---|
| AlexNet | *76.1 ± 3.8* | 72.8 ± 2.7 | *75.7 ± 4.0* | **76.8 ± 3.6** | 63.9 ± 12.5 | *75.9 ± 3.5* | 74.9 ± 3.9 |
| GoogLeNet | *79.9 ± 1.8* | *78.3 ± 2.6* | 77.3 ± 3.1 | *79.2 ± 4.0* | 69.4 ± 12.0 | **80.0 ± 3.8** | *78.4 ± 4.0* |
| MobileNet_v2 | **81.8 ± 4.3** | 80.7 ± 2.5 | 78.8 ± 3.4 | 81.2 ± 4.9 | 79.8 ± 3.7 | 79.2 ± 3.2 | 80.8 ± 3.7 |
| ResNet18 | 79.8 ± 2.6 | 77.2 ± 2.3 | 78.5 ± 4.1 | **80.7 ± 4.1** | 75.2 ± 8.4 | *80.4 ± 3.8* | 78.0 ± 4.3 |
| ResNeXt50_32x4d | *82.0 ± 3.1* | 80.0 ± 3.5 | 79.5 ± 3.2 | **82.3 ± 4.3** | 76.2 ± 5.1 | 80.8 ± 2.8 | 79.9 ± 3.9 |
| ShuffleNet_v2_x1_0 | **77.1 ± 3.7** | 72.1 ± 3.5 | *76.1 ± 3.5* | 75.0 ± 4.4 | 70.4 ± 6.6 | *76.9 ± 3.9* | 76.2 ± 2.3 |
| SqueezeNet1_0 | *77.2 ± 4.2* | 73.5 ± 3.1 | 74.9 ± 5.1 | **77.3 ± 5.3** | 49.9 ± 12.2 | 75.5 ± 3.3 | 74.3 ± 4.5 |
| VGG16 | *77.9 ± 4.8* | 74.4 ± 4.7 | *77.5 ± 3.8* | **79.4 ± 2.5** | 58.1 ± 11.8 | 77.0 ± 3.9 | *77.4 ± 3.7* |
| Wide_ResNet50_2 | 80.8 ± 3.2 | 79.3 ± 3.3 | **82.2 ± 4.2** | *81.0 ± 3.9* | 64.0 ± 15.3 | *81.3 ± 4.2* | 80.6 ± 2.6 |
| **Avg** | 79.2 | 76.5 | 77.8 | 79.2 | 67.4 | 78.5 | 77.8 |
| **Winners** | 2 | 0 | 1 | 5 | 0 | 1 | 0 |

Notes.
**Bold**, best model, *italic:* statistically similar to best (paired *t*-test).

**Table 8  Results in terms of the Uniform Ordinal Classification Index (UOC) for four class problem, averaged for 10 folds (lower is better).**

|  | CE | BU | PU | OE | CO | CO2 | HO2 |
|---|---|---|---|---|---|---|---|
| AlexNet | *38.2 ± 5.1* | 39.5 ± 3.4 | *37.1 ± 4.3* | **37.0 ± 4.9** | 52.7 ± 14.2 | *37.4 ± 5.8* | *38.9 ± 5.8* |
| GoogLeNet | *31.6 ± 3.1* | *31.7 ± 3.6* | 34.4 ± 5.6 | *32.5 ± 5.7* | 44.7 ± 14.6 | **30.8 ± 5.5** | *32.9 ± 6.3* |
| MobileNet_v2 | *30.1 ± 6.9* | **29.2 ± 3.7** | 32.8 ± 5.2 | *30.6 ± 7.5* | *31.0 ± 4.8* | 32.5 ± 5.5 | *30.5 ± 5.4* |
| ResNet18 | 31.4 ± 4.6 | 33.1 ± 3.7 | 32.3 ± 5.5 | **29.4 ± 6.0** | 36.7 ± 11.0 | *30.3 ± 4.1* | 33.2 ± 6.7 |
| ResNeXt50_32x4d | *28.7 ± 4.7* | *29.8 ± 4.9* | 32.0 ± 3.9 | **27.5 ± 5.3** | 35.9 ± 4.8 | *28.8 ± 4.6* | 31.0 ± 5.2 |
| ShuffleNet_v2_x1_0 | **35.8 ± 5.3** | 38.6 ± 4.7 | 36.7 ± 4.4 | 39.0 ± 6.5 | 43.5 ± 9.0 | *36.4 ± 6.9* | 35.9 ± 4.7 |
| SqueezeNet1_0 | *36.6 ± 5.8* | 37.3 ± 4.3 | 38.2 ± 6.8 | **35.3 ± 6.9** | 65.1 ± 9.4 | 37.6 ± 4.1 | 39.6 ± 4.6 |
| VGG16 | 35.3 ± 6.4 | 36.2 ± 6.4 | *34.6 ± 4.7* | **32.3 ± 3.8** | 55.1 ± 10.5 | 34.7 ± 5.5 | 35.1 ± 6.0 |
| Wide_ResNet50_2 | 30.2 ± 5.7 | *29.9 ± 4.9* | **28.2 ± 5.0** | *30.5 ± 6.2* | 47.7 ± 14.4 | *29.1 ± 5.6* | 30.7 ± 4.3 |
| **Avg** | 33.1 | 33.9 | 34.0 | 32.7 | 45.8 | 33.1 | 34.2 |
| **Winners** | 1 | 1 | 1 | 5 | 0 | 1 | 0 |

Notes.
**Bold**, best model, *italic:* statistically similar to best (paired *t*-test).

**Table 9  Accuracy comparison of different models with literature for seven and four classes.**

|  | 7 classes Accuracy (%) | 4 classes Accuracy (%) |
|---|---|---|
| Jantzen et al. | 61.1 | – |
| Lin et al. | 64.5 | 71.3 |
| Proposal | **75.6** | **81.3** |

kinds of cells are very similar. The fact is that the abnormal classes correspond to different levels of evolution of structures, with a progressive change in their characteristics which leads them to present characteristics common to two levels, being a hard task even for cytopathologists to classify them correctly. Thus, the right multi-class classification of abnormal cells is highly desirable and with substantial clinical value.

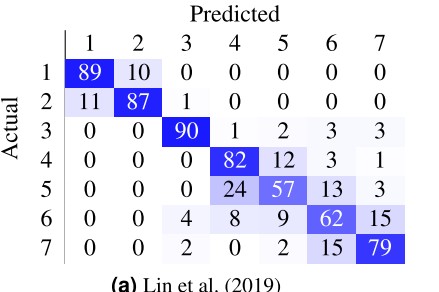
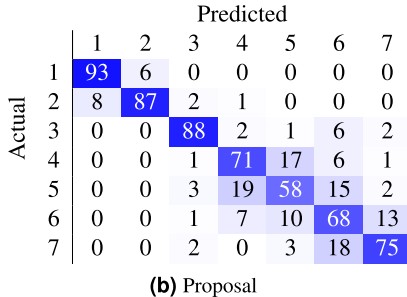

**Figure 5  Comparison of state-of-the-art confusion matrix (seven classes) against WideResNet50 trained using the HO2 loss.** (A) *Lin et al. (2019)*; (B) proposal.

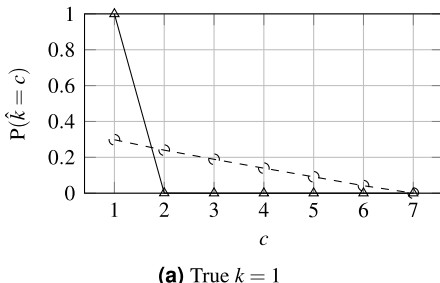
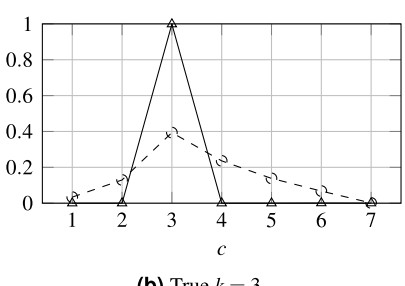

**Figure 6  Probability distribution for WideResNet50 contrasting losses CE (solid line) and HO2 (dashed line).** (A) True $k = 1$; (B) True $k = 3$.

Finally, the influence of the losses on the output probabilities is illustrated in Fig. 6 when predicting two classes for the 7-class case. Contrasting this to Fig. 1, it is clear that the proposed loss tends to promote a unimodal distribution of probabilities relative to the CE loss, which tends to maximize the probability in the output corresponding to the true class and ignore all the other probabilities distribution, and even in contrast to OE. Concerning the sparsity of the prediction probabilities, as measured by the Gini index, it is notable that, as the loss is made more ordinal-aware, the predicted probabilities tend to be more spread across the classes. This can also be seen in Fig. 6. Interestingly, the OE distribution is almost identical to the CE distribution and has been omitted from the figure for legibility.

## CONCLUSION

Comparing ordinal deep learning approaches on cervical cancer data, non-parametric losses achieved better results when comparing with parametric losses. This type of loss does not limit the learned representation to a specific parametric model, which allows, during the training, to explore different and larger spaces of solutions avoiding ad hoc choices.

A new non-parametric loss is proposed for multi-class Pap smear cell-classification based on convolutional neural networks. This new loss demonstrated to be competitive with state-of-the-art results and more flexible than existing deep ordinal classification techniques

that impose uni-modality in the probability distribution. The use of the proposed loss in training popular architectures from the literature outperforms the state-of-the-art nominal-class approaches by over 10%.

Furthermore, the proposed loss is a convenient way of introducing ordinality to the optimization problem without the major changes in architecture or data format required by other techniques from the literature. On the other hand, the proposed loss requires two new hyperparameters. However, the suggested values have been found to be robust. While motivated by this dataset, the proposed loss could potentially be used by other applications of ordinal classification.

In any case, there is a lot to improve in the multi-class classification of cervical cells to achieve better accuracy since results are still short of 75.6% accuracy. The Herlev data set is mainly composed of expert-selected "typical" cells, however, in real-life circumstances, data is more complex because a cytology image contains lots of cells and not only a single cropped cell, so further work is needed before moving the results of this work to practice. Another important detail is the effect of overlapping nuclei and cell clumps, which has not been taken into account in this work. The presence of artefacts on the images also interferes with classification accuracy.

## ACKNOWLEDGEMENTS

The authors would like to acknowledge access to the Herlev Pap smear dataset collected by Herlev University Hospital (Denmark) and the Technical University of Denmark.

## APPENDIX

Some extra results are made available in this appendix.

**Table A1  Results in terms of Kendall's $\tau$ for seven class problem, averaged for 10 folds (higher is better).**

|  | CE | BU | PU | OE | CO | CO2 | HO2 |
|---|---|---|---|---|---|---|---|
| AlexNet | 75.9 ± 4.1 | 76.9 ± 4.8 | 74.5 ± 5.8 | *77.9 ± 4.8* | 57.3 ± 9.3 | **79.9 ± 4.7** | 77.5 ± 5.5 |
| GoogLeNet | *80.9 ± 2.7* | *81.9 ± 4.1* | 79.3 ± 4.4 | *81.9 ± 4.3* | 76.9 ± 4.0 | *81.7 ± 4.2* | **82.8 ± 3.7** |
| MobileNet_v2 | 83.2 ± 2.9 | 84.5 ± 1.9 | **85.6 ± 2.5** | *84.4 ± 2.1* | 75.6 ± 11.5 | 83.7 ± 3.2 | 83.8 ± 3.0 |
| ResNet18 | *83.1 ± 5.1* | **84.7 ± 3.1** | 83.4 ± 3.0 | *83.5 ± 4.9* | 76.9 ± 5.1 | *83.8 ± 4.0* | 83.4 ± 5.1 |
| ResNeXt50_32x4d | 83.2 ± 4.5 | *85.2 ± 2.8* | 84.7 ± 1.4 | 83.2 ± 3.6 | 81.5 ± 3.4 | *84.0 ± 3.6* | **85.4 ± 3.1** |
| ShuffleNet_v2_x1_0 | 78.8 ± 4.0 | 78.9 ± 3.6 | 80.9 ± 2.4 | **81.7 ± 3.1** | 77.8 ± 4.5 | 80.1 ± 2.6 | *81.6 ± 3.4* |
| SqueezeNet1_0 | **81.0 ± 3.9** | 80.4 ± 2.6 | 77.0 ± 4.0 | *79.6 ± 5.1* | 54.1 ± 14.6 | *79.4 ± 4.1* | 78.1 ± 4.8 |
| VGG16 | *81.6 ± 5.1* | 81.3 ± 2.2 | 78.3 ± 6.4 | *81.9 ± 3.5* | 68.8 ± 6.9 | *82.6 ± 3.2* | **82.8 ± 4.6** |
| Wide_ResNet50_2 | 83.4 ± 3.1 | 83.8 ± 2.8 | 84.9 ± 3.4 | **85.9 ± 2.7** | 79.8 ± 6.1 | 84.1 ± 3.6 | 82.7 ± 4.3 |
| **Avg** | 81.2 | 82.0 | 80.9 | 82.2 | 72.1 | 82.2 | 82.0 |
| **Winners** | 1 | 1 | 1 | 2 | 0 | 1 | 3 |

Notes.
  **Bold:** best model, *italic:* statistically similar to best (paired *t*-test).

**Table A2  Results in terms of Kendall's $\tau$ for 4 class problem, averaged for 10 folds (higher is better).**

|  | CE | BU | PU | OE | CO | CO2 | HO2 |
|---|---|---|---|---|---|---|---|
| AlexNet | 73.9 ± 6.3 | 75.4 ± 4.0 | **77.4 ± 3.8** | *76.0 ± 5.7* | 58.7 ± 19.3 | *76.5 ± 5.7* | 75.0 ± 5.9 |
| GoogLeNet | 80.3 ± 4.2 | *81.3 ± 3.0* | 80.4 ± 4.9 | *81.2 ± 4.4* | 70.1 ± 14.2 | **82.5 ± 4.2** | 80.2 ± 5.1 |
| MobileNet_v2 | 81.3 ± 5.1 | **83.9 ± 2.0** | 81.3 ± 3.7 | *81.8 ± 6.2* | 82.3 ± 3.7 | 80.4 ± 4.3 | 82.3 ± 4.4 |
| ResNet18 | 81.0 ± 2.7 | 80.3 ± 4.4 | *81.2 ± 4.4* | **82.8 ± 3.5** | 77.5 ± 7.7 | *82.5 ± 3.2* | 79.5 ± 5.1 |
| ResNeXt50_32x4d | *83.2 ± 3.1* | 83.2 ± 2.9 | 81.8 ± 2.8 | **84.3 ± 3.4** | 78.1 ± 5.1 | *83.9 ± 2.4* | 82.3 ± 3.9 |
| ShuffleNet_v2_x1_0 | 77.1 ± 4.7 | 76.1 ± 3.7 | **78.8 ± 4.1** | 74.8 ± 6.2 | 71.1 ± 9.1 | *77.3 ± 5.6* | *78.0 ± 4.5* |
| SqueezeNet1_0 | *76.2 ± 5.2* | 77.5 ± 4.5 | 75.8 ± 5.6 | **78.2 ± 5.9** | 48.5 ± 11.0 | *76.6 ± 3.8* | 74.8 ± 3.8 |
| VGG16 | 77.9 ± 5.4 | 79.2 ± 4.5 | 80.0 ± 4.4 | **81.0 ± 2.4** | 63.2 ± 10.3 | 79.2 ± 4.2 | 78.3 ± 5.4 |
| Wide_ResNet50_2 | 81.7 ± 4.6 | 83.4 ± 2.8 | **84.4 ± 4.6** | *82.2 ± 5.1* | 67.8 ± 18.6 | *83.2 ± 4.3* | *82.6 ± 2.8* |
| **Avg** | 79.2 | 80.0 | 80.1 | 80.2 | 68.6 | 80.3 | 79.2 |
| **Winners** | 0 | 1 | 3 | 4 | 0 | 1 | 0 |

Notes.
  **Bold:** best model, *italic:* statistically similar to best (paired *t*-test).

**Table A3  Aggregate results for 7 class problem, averaged for 10 folds.**

|  | CE | BU | PU | OE | CO | CO2 | HO2 |
|---|---|---|---|---|---|---|---|
|  | | | | **Mode** | | | |
| UOC | *38.8 ± 7.5* | 44.2 ± 7.2 | 41.1 ± 7.7 | **38.6 ± 7.5** | 53.5 ± 14.7 | *39.0 ± 7.3* | 39.5 ± 8.1 |
| MAE | 0.37 ± 0.08 | 0.41 ± 0.09 | 0.40 ± 0.10 | **0.36 ± 0.08** | 0.60 ± 0.26 | *0.37 ± 0.07* | *0.37 ± 0.09* |
| Accuracy | **73.4 ± 4.8** | 66.2 ± 5.5 | 71.5 ± 4.9 | 72.8 ± 5.1 | 58.6 ± 13.4 | 72.6 ± 4.5 | 72.0 ± 5.6 |
| Kendall's $\tau$ | 81.2 ± 4.7 | *82.0 ± 4.2* | 80.0 ± 5.8 | **82.2 ± 4.5** | 72.1 ± 12.4 | *82.2 ± 4.2* | *82.0 ± 4.9* |
| ROC AUC | **95.9 ± 1.4** | 93.0 ± 2.2 | 95.5 ± 1.4 | 95.5 ± 1.5 | 82.3 ± 9.9 | 94.5 ± 1.8 | 93.9 ± 1.9 |
| Gini | 85.1 ± 0.2 | 64.0 ± 1.6 | 84.8 ± 0.5 | 84.8 ± 0.4 | 28.2 ± 33.3 | 50.1 ± 6.3 | 45.0 ± 4.2 |
|  | | | | **Mean** | | | |
| UOC | **39.2 ± 7.4** | 42.1 ± 7.5 | 41.7 ± 8.2 | *39.7 ± 7.4* | 75.9 ± 20.6 | 79.8 ± 4.9 | 83.7 ± 1.7 |
| MAE | *0.37 ± 0.08* | 0.39 ± 0.08 | 0.39 ± 0.09 | **0.37 ± 0.08** | 1.17 ± 0.50 | 0.91 ± 0.13 | 1.03 ± 0.04 |
| Accuracy | **72.1 ± 5.1** | 67.7 ± 5.8 | 70.1 ± 5.6 | *71.6 ± 5.1* | 33.6 ± 19.9 | 28.3 ± 5.4 | 25.0 ± 3.1 |
| Kendall's $\tau$ | *82.3 ± 4.3* | 82.5 ± 4.1 | 80.9 ± 5.5 | **82.6 ± 4.3** | *nan ± nan* | 78.3 ± 3.6 | 76.0 ± 3.2 |
| ROC AUC | **95.9 ± 1.4** | 93.0 ± 2.2 | 95.5 ± 1.4 | 95.5 ± 1.5 | 82.3 ± 9.9 | 94.5 ± 1.8 | 93.9 ± 1.9 |
| Gini | 85.1 ± 0.2 | 64.0 ± 1.6 | 84.8 ± 0.5 | 84.8 ± 0.4 | 28.2 ± 33.3 | 50.1 ± 6.3 | 45.0 ± 4.2 |

Notes.
  **Bold:** best model, *italic:* statistically similar to best (paired *t*-test).

**Table A4  Aggregate results for 4 class problem, averaged for 10 folds.**

|  | CE | BU | PU | OE | CO | CO2 | HO2 |
|---|---|---|---|---|---|---|---|
|  | | | | **Mode** | | | |
| UOC | *33.1 ± 6.3* | 33.9 ± 5.9 | 33.6 ± 5.7 | **32.7 ± 7.0** | 45.8 ± 14.9 | *33.1 ± 6.3* | 34.2 ± 6.4 |
| MAE | 0.25 ± 0.06 | 0.27 ± 0.06 | *0.25 ± 0.06* | **0.24 ± 0.06** | 0.40 ± 0.20 | *0.25 ± 0.06* | 0.26 ± 0.06 |
| Accuracy | *79.2 ± 4.1* | 76.5 ± 4.5 | 78.7 ± 4.0 | **79.2 ± 4.8** | 67.4 ± 13.7 | 78.5 ± 4.2 | 77.8 ± 4.3 |
| Kendall's $\tau$ | 79.2 ± 5.5 | 80.0 ± 4.7 | 79.7 ± 5.4 | 80.2 ± 5.8 | 68.6 ± 15.7 | **80.3 ± 5.1** | 79.2 ± 5.4 |
| ROC AUC | 94.5 ± 1.7 | 92.7 ± 1.9 | **94.6 ± 1.5** | *94.5 ± 1.7* | 83.5 ± 10.5 | 93.2 ± 2.1 | 92.9 ± 2.0 |
| Gini | 74.3 ± 0.3 | 58.4 ± 2.0 | 74.2 ± 0.4 | 74.2 ± 0.3 | 31.5 ± 31.4 | 44.5 ± 16.3 | 36.6 ± 11.2 |

**Table A4** (*continued*)

|  | CE | BU | PU | OE | CO | CO2 | HO2 |
|---|---|---|---|---|---|---|---|
|  |  |  |  | Mean |  |  |  |
| UOC | *33.4 ± 6.3* | 34.2 ± 5.8 | *34.0 ± 5.9* | **33.2 ± 7.1** | 57.0 ± 17.3 | 52.2 ± 13.5 | 58.4 ± 8.5 |
| MAE | *0.25 ± 0.06* | 0.26 ± 0.05 | *0.25 ± 0.06* | **0.25 ± 0.07** | 0.50 ± 0.20 | 0.42 ± 0.14 | 0.47 ± 0.09 |
| Accuracy | **78.5 ± 4.4** | 77.0 ± 4.2 | 77.8 ± 4.4 | *78.4 ± 5.1* | 54.7 ± 17.8 | 60.1 ± 13.4 | 54.6 ± 9.3 |
| Kendall's $\tau$ | *79.7 ± 5.1* | *80.4 ± 4.6* | *80.1 ± 4.9* | **80.5 ± 5.8** | 63.8 ± 16.6 | 73.4 ± 7.3 | 70.5 ± 5.6 |
| ROC AUC | *94.5 ± 1.7* | 92.7 ± 1.9 | **94.6 ± 1.5** | *94.5 ± 1.7* | 83.5 ± 10.5 | 93.2 ± 2.1 | 92.9 ± 2.0 |
| Gini | 74.3 ± 0.3 | 58.4 ± 2.0 | 74.2 ± 0.4 | 74.2 ± 0.3 | 31.5 ± 31.4 | 44.5 ± 16.3 | 36.6 ± 11.2 |

**Notes.**
    **Bold:** best model, *italic:* statistically similar to best (paired *t*-test).

### Funding

The project TAMI-Transparent Artificial Medical Intelligence (NORTE-01-0247-FEDER-045905) funding this work is co-financed by ERDF - European Regional Fund through the North Portugal Regional Operational Program-NORTE 2020 and by the Portuguese Foundation for Science and Technology-FCT under the CMU-Portugal International Partnership. Ricardo Cruz was supported by Ph.D. grant SFRH/BD/122248/2016, also provided by FCT. There was no additional external funding received for this study. The funders had no role in study design, data collection and analysis, decision to publish, or preparation of the manuscript.

### Grant Disclosures

The following grant information was disclosed by the authors:
ERDF – European Regional Development Fund.
Portuguese Foundation for Science and Technology-FCT.
FCT: SFRH/BD/122248/2016.

### Competing Interests

The authors declare there are no competing interests.

### Author Contributions

- Tomé Albuquerque, Ricardo Cruz and Jaime Cardoso conceived and designed the experiments, performed the experiments, analyzed the data, performed the computation work, prepared figures and/or tables, authored or reviewed drafts of the paper, and approved the final draft.

### Data Availability

    Data (''smear2005.zip) is available at MDE-Lab:
    J. Jantzen, G. Dounias, Analysis of Pap-Smear Data. NISIS 2006, Puerto de la Cruz, Tenerife, Spain, 2006. http://mde-lab.aegean.gr/index.php/downloads.
    Code is available at GitHub: https://github.com/tomealbuquerque/ordinal-losses.

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
