# Peer review of "Ordinal losses for classification of cervical cancer risk"

_PeerJ Computer Science, doi:10.7717/peerj-cs.457_

## Round 0.1 · original submission · Major Revisions

· Academic Editor

Major Revisions

The experimental evaluation and comparison need strengthening based on Reviewer 2 and 3's comments.

Regarding Reviewer 3's comments, given the focus on cervical cancer imaging, it may not be possible or necessary to extend the collection of datasets used in the study. However, significance tests need to be performed on the estimates of predictive performance, and measures of variance need to be provided. This should be feasible even with deep learning given the small size of the dataset.

The exact experimental protocol used in the study remains unclear to me. It is important to state exactly how much data is used for training, validation (i.e., parameter tuning and early stopping), and testing. If stratified k-fold cross-validation is performed to establish the final performance estimates, then parameter tuning needs to be performed separately for each of the k runs, making sure that information from the test set of run k does not influence hyperparameter choice for run k in any way.

In my opinion, given the current results, where OE appears very competitive with the proposed new ordinal loss functions, the paper should deemphasise the novel loss functions and instead focus on the possibility that using ordinal methods improves results on this cancer data (assuming superiority holds after significance testing).

An empirical comparison of different deep ordinal classification approaches (including the new ones) on this data seems a valuable contribution. In this regard, the suggestions by Reviewer 2 need to be addressed, particularly the simple baseline using the "expectation trick" and the published deep ordinal methods cited in the review.

·

Basic reporting

The writing is unambiguous and easy to follow. Background and related work are clear and rather detailed. "Fully automatic knee osteoarthritis severity grading using deep neural networks with a novel ordinal loss" is suggested to add to the related work.

Experimental design

Considering the ordinal nature of the pap smear cell classification, the authors propose a non-parametric ordinal loss to promote the output probabilities to follow a unimodal distribution.

Validity of the findings

The authors experiment with the proposed loss on the Herlev dataset on multiple CNN architectures. In addition, the authors compare several other losses. Experiments show the effectiveness of the proposed methods.

Additional comments

The authors propose a novel ordinal loss for the pap smear cell classification. They focus on promoting the unimodal distribution of the output probabilities, which is a good insight into the ordinal classification problem. The experiments and evaluations well demonstrate the idea.

·

Basic reporting

no comment

Experimental design

no comment

Validity of the findings

The authors have presented an extensive empirical evaluation of their proposed method, as well as several baselines. While the results are indeed decent, it is not clear to me whether the complexity of the method would hold up to some stronger baselines (more on this soon). For example, the proposed loss is a summation of many piece-wise terms to enforce unimodality (linear in the number of classes) and involves two hyperparameters, whereas it seems to me like there are much simpler ways one could enforce 'semi-unimodality'.

To elaborate on my 'stronger baselines' point, it seems like the main reason why this loss was proposed is because we do not necessarily want a distribution that is purely unimodal (like in the case of PU). Perhaps that is partly because (1) the conditional probability distribution should not be modelled by a unimodal distribution; and/or (2) using PU (i.e. a binomial distribution) would be too constraining since the variance cannot be easily controlled.

To address point (2): a variance-controlled version of the binomial distribution does exist -- called the Conway-Maxwell Binomial (CMB) [2,3] -- which has a variance-controlling term. That means that your network could be modified to have two outputs: (p, v) = f(x), and then you can maximise the log-likelihood of a CMB distribution. (A more heuristic version of this was proposed in [1], but it's essentially CMB.)

Secondly, to address point (1): why not just infer a mixture distribution between a regular softmax (CE) distribution and a unimodal (PU) one? For instance, suppose your network was modified to have two outputs: p_s(y|x) and p_u(y|x), where p_s denotes a regular softmax distribution and p_u the unimodal one, you could simply (for some alpha in [0,1]) construct a mixture distribution between the two: p(y|x) = alpha*p_u(y|x) + (1-alpha)*p_s(y|x). alpha could either be a hyperparameter to tune, or you might even be able to get away with making it a learnable parameter as part of the network. This would make for a somewhat interesting method, since a high value of alpha would put more weight on p_u, essentially acting as a strong regulariser on the distribution.

Thirdly, the more competitive version of the simplest baseline (CE) would be to do a post-hoc label prediction based on the 'expectation trick' found in [1]. Essentially, for some p(y|x), if we assign to each class an integer label [1, 2, ..., K], we take the expected value of this distribution by computing the dot product between p(y|x) and [1, 2, ... K], and round the result to the nearest whole integer. This basically uses all of the probability mass in p(y|x) to make a prediction.

In summary, I would compare your proposed technique to:
- A more competitive CE using the expectation trick
- Use a parametric unimodal (PU) method using the CMB distribution
- Experiment with using a mixture distribution between p_u and p_s

[4] also proposes a solution to (2), using the Wasserstein distance. [5] may also be worth reading.
* * *
Other less significant points:
- While it was appreciated that the authors tried out a vast range of architectures, perhaps it would make for a better presentation if the number of learnable parameters was stated for each of these architectures. You could then explore performance vs # parameters. It seems like the dataset you have used is extremely tiny, and having excessively large networks could degrade generalisation performance here. If it saves you computational resources, I don't think some of these architectures are strictly needed in the analysis: for instance AlexNet and VGG, which were superceded by ResNets (and for good reason).
- It would be interesting to explore the case where you don't start off with pre-trained ImageNet weights. While I would expect such a network to very easily overfit, it can be controlled with sufficient regularisation (weight decay), and also allow you to explore the effect of having a severely constrained distribution (i.e. PU) in a 'low data' setting.

[1] Beckham, C. & Pal, C.. (2017). Unimodal Probability Distributions for Deep Ordinal Classification. Proceedings of the 34th International Conference on Machine Learning, in PMLR 70:411-419

[2]: https://en.wikipedia.org/wiki/Conway%E2%80%93Maxwell%E2%80%93binomial_distribution

[3]: https://github.com/christopher-beckham/conway-maxwell-binomial/blob/master/implementation.ipynb

[4]: Hou, L., Yu, C. P., & Samaras, D. (2016). Squared earth mover's distance-based loss for training deep neural networks. arXiv preprint arXiv:1611.05916.

[5]: Liu, X., Fan, F., Kong, L., Diao, Z., Xie, W., Lu, J., & You, J. (2020). Unimodal regularized neuron stick-breaking for ordinal classification. Neurocomputing.

Reviewer 3 ·

Basic reporting

no comment

Experimental design

Only averages of 5 folds are given, in order to show the robustness I suggest to provide results of multiple experiments (e.g. 10), then averages and variance / standard deviation or box plots. And in addition I suggest to perform statistical significance tests on the prosed and tested algorithms.

Validity of the findings

The proposed algorithm cost functions are straightforward. It would be a surprise if ordinal classification can benefit from these cost functions in general. Only a rigorous statistical evaluation of the proposed cost functions based on 10 or more data sets utilising more complex statistical evaluation (e.g. Wilcoxons test) in comparison with the other approaches could prove the strength of the proposed algorithm.

Additional comments

see Part 2 and Part 3.

---

## Round 0.2 · Major Revisions

· Academic Editor

Major Revisions

Thank you very much for your efforts to improve the paper. Reviewer 2 has some comments regarding the changes that you have made. Please address these (which should hopefully not require a substantial amount of work) and resubmit.

·

Basic reporting

no comment

Experimental design

no comment

Validity of the findings

no comment

Additional comments

Revision well resolved raised issues.

·

Basic reporting

Regarding the unimodal papers cited: currently [2] is called "Poisson unimodal" (PU). This is a mischaracterisation because [2] also explores the binomial distribution. Furthermore, although [1] only considers the binomial distribution (BU), [3] (same authors as [1]) wrote a more recent paper which considers both binomial and Poisson. Therefore, I would suggest that you simply refer to both styles (binomial/poisson) as simply "Unimodal" and cite [2] and [3] together. E.g. from lines 133-147 you simply say something like "Unimodal (da Costa et al (2008), Beckham & Pal (2017)) constrains the output of ..."

Also, the difference between [2] and [3] is simply that [2] is exploring binomial/poisson in the context of deep learning (rather than classical ML), [2] also proposes using a learnable softmax temperature term to control the variance of the distribution (which I mentioned in my first review as the Conway-Maxwell {Binomial,Poisson} distribution). Therefore, in lines 141-147 you should explicitly state whether you are learning or tuning such a temperature term or not. If you're not doing this, then your implementation of PU/BU is going to be more faithful to [3] than it is to [2], and that should be mentioned.

[1]: Da Costa, J. P., & Cardoso, J. S. (2005, October). Classification of ordinal data using neural networks. In European Conference on Machine Learning (pp. 690-697). Springer, Berlin, Heidelberg.
[2]: Beckham, C., & Pal, C. (2017). Unimodal probability distributions for deep ordinal classification. arXiv preprint arXiv:1705.05278.
[3]: da Costa, J. F. P., Alonso, H., & Cardoso, J. S. (2008). The unimodal model for the classification of ordinal data. Neural Networks, 21(1), 78-91.
* * *
I appreciate that uncertainty estimates have been added to the experiments, though now I have various issues with how they are being presented.

In line 271 you say "In all the Tables of results, the best results for each loss are in bold", followed by "Furthermore, the other results are compared to the best result and a statistical test is used with the hypothesis of them being the same or not". This makes it sound like the bold numbers are simply referring to the "best" value (i.e. lowest error, or higher accuracy), with statistical testing being something else added to the table. When I look at the table, for each row (with each row denoting an architecture), I see multiple bold results. The issue here is that, in most ML papers the bold number traditionally means the "best" value, whereas here I am seeing that it is referring to a whether that result is statistically significant or not. But to make matters even more confusing, the statistical test seems to be done with respect to the *best* loss. For instance, for each row, rather than doing t_test(baseline, loss1), t_test(baseline, loss2), etc... it looks like you are doing t_test(loss1, best_loss), t_test(loss2, best_loss), etc. It would be much easier to interpret the result if each t-test was performed *with respect* to the CE baseline.

I would suggest the following changes to the table:
- Instead of using boldface to determine statistical significance, format the table similar to how it is done in [4]'s Table 2, i.e. a black dot next to the number (x +/- y) indicates whether that result is statistically significant *with respect* to the baseline (CE) loss.
- To also be consistent with most deep learning papers, use boldface to highlight the *best* result (i.e. lowest MAE, or highest accuracy, etc.). You could constrain the "best" result to only be amongst the ones that passed the t-test. This means that for each row in the table, only one of the losses is going to be in bold.

You also need to make it clear in the text if these table numbers are averages from the validation folds or test folds I am assuming that in your k-fold cross-validation setup, for each iteration, 1/k folds is the valid set (for tuning HPs) and another 1/k folds is the 'test set'. That means that when the k-fold cross-validation is completed you will have results for a total of k validation folds and k test folds. Can you confirm that this is the case? (Make this clear in lines 228-236.) *

(*Some experimental setups may opt to have a held-out test set that is completely isolated from the cross-validation loop, but I assume you are not doing this because of how little data you have.)

[4]: Frank, E., & Hall, M. (2001, September). A simple approach to ordinal classification. In European Conference on Machine Learning (pp. 145-156). Springer, Berlin, Heidelberg.
* * *
Minor comments:

- I would have preferred if the expectation trick result was made a part of the main text rather than relegated to an appendix table, but I won't make a big deal out of this.
* * *
Typos, etc:

- Figure 5a: Lin et al is mentioned twice
- Line 204: "512 dense layer" --> "512-unit dense layer", same for 256 dense layer

Experimental design

See my comment in Section 1 about being more clear in the text about how the cross-validation is being performed.

Validity of the findings

no comment

Additional comments

no comment

---

## Author Rebuttal · Round 0.2

December 9, 2020

INESC TEC
Campus da Faculdade de Engenharia da Universidade do Porto
Rua Dr. Roberto Frias
4200-465 Porto
Portugal

**Article id**: 53280
**Article Title**: Ordinal Losses for Classification of Cervical Cancer Risk

Dear Editors,

We thank the reviewers for their generous comments on the manuscript and we have edited the manuscript to address their concerns.

All of the code we wrote is available and I have included the link throughout the paper to the appropriate code repository.

We look forward to hear what your thoughts are about the updated manuscript.

Yours sincerely,

Tomé Mendes Albuquerque

On behalf of all authors.

**Editor:**

Regarding Reviewer 3's comments, given the focus on cervical cancer imaging, it may not be possible or necessary to extend the collection of datasets used in the study. However, significance tests need to be performed on the estimates of predictive performance, and measures of variance need to be provided. This should be feasible even with deep learning given the small size of the dataset.

The exact experimental protocol used in the study remains unclear to me. It is important to state exactly how much data is used for training, validation (i.e., parameter tuning and early stopping), and testing. If stratified k-fold cross-validation is performed to establish the final performance estimates, then parameter tuning needs to be performed separately for each of the k runs, making sure that information from the test set of run k does not influence hyperparameter choice for run k in any way.

We agree with your concern and we implemented stratified 10-fold cross-validation and we also implemented nested k-fold to do parameter tuning ($\lambda$ of our proposal loss). We updated the manuscript including in "Train" subsection more information about cross-validation methodologies used during the train.

In my opinion, given the current results, where OE appears very competitive with the proposed new ordinal loss functions, the paper should deemphasise the novel loss functions and instead focus on the possibility that using ordinal methods improves results on this cancer data (assuming superiority holds after significance testing).

An empirical comparison of different deep ordinal classification approaches (including the new ones) on this data seems a valuable contribution. In this regard, the suggestions by Reviewer 2 need to be addressed, particularly the simple baseline using the "expectation trick" and the published deep ordinal methods cited in the review.

We agree with your concern that we should deemphasise the novel loss functions and focus on the idea that using ordinal methods improves results on this cancer data. We updated the manuscript by changing article title to "Ordinal Losses for Classification of Cervical Cancer Risk" and by adding more details between parametric and no-parametric losses. We also write in "results" and "conclusion" sections a critical analysis of the results obtained by nominal losses (CE) VS ordinal losses and we also analyze the differences between parametric and non-parametric losses in relation to their performance.

**Reviewer 1: Pingjun Chen**

**Basic reporting:** The writing is unambiguous and easy to follow. Background and related work are clear and rather detailed. "Fully automatic knee osteoarthritis severity grading using deep neural networks with a novel ordinal loss" is suggested to add to the related work.

**Experimental design:** Considering the ordinal nature of the pap smear cell classification, the authors propose a non-parametric ordinal loss to promote the output probabilities to follow a unimodal distribution.

**Validity of the findings:** The authors experiment with the proposed loss on the Herlev dataset on multiple CNN architectures. In addition, the authors compare several other losses. Experiments show the effectiveness of the proposed methods.

**Comments for the author:** The authors propose a novel ordinal loss for the pap smear cell classification. They focus on promoting the unimodal distribution of the output probabilities, which is a good insight into the ordinal classification problem. The experiments and evaluations well demonstrate the idea.

**Reviewer 2: Christopher Beckham**

To elaborate on my 'stronger baselines' point, it seems like the main reason why this loss was proposed is because we do not necessarily want a distribution that is purely unimodal (like in the case of PU). Perhaps that is partly because (1) the conditional probability distribution should not be modelled by a unimodal distribution; and/or (2) using PU (i.e. a binomial distribution) would be too constraining since the variance cannot be easily controlled.

To address point (2): a variance-controlled version of the binomial distribution does exist – called the Conway-Maxwell Binomial (CMB) [2,3] – which has a variance-controlling term. That means that your network could be modified to have two outputs: (p, v) = f(x), and then you can maximise the log-likelihood of a CMB distribution. (A more heuristic version of this was proposed in [1], but it's essentially CMB.)

Secondly, to address point (1): why not just infer a mixture distribution between a regular softmax (CE) distribution and a unimodal (PU) one? For instance, suppose your network was modified to have two outputs: $p\_s(y|x) and p\_u(y|x)$, where $p\_s$ denotes a regular softmax distribution and p_u the unimodal one, you could simply (for some alpha in [0,1]) construct a mixture distribution between the two: $p(y|x) = alpha * p\_u(y|x) + (1 - alpha) * p\_s(y|x)$. alpha could either be a hyperparameter to tune, or you might even be able to get away with making it a learnable parameter as part of the network. This would make for a somewhat interesting method, since a high value of alpha would put more weight on $p\_u$, essentially acting as a strong regulariser on the distribution.

Thirdly, the more competitive version of the simplest baseline (CE) would be to do a post-hoc label prediction based on the 'expectation trick' found in [1]. Essentially, for some $p(y|x)$, if we assign to each class an integer label $[1, 2, \ldots, K]$, we take the expected value of this distribution by computing the dot product between $p(y|x)$ and $[1, 2, \ldots, K]$, and round the result to the nearest whole integer. This basically uses all of the probability mass in $p(y|x)$ to make a prediction.

In summary, I would compare your proposed technique to: - A more competitive CE using the expectation trick

We agree with your concern and we implemented the expectation trick for all losses and architectures. We updated the manuscript by adding two new tables (Table A3. and A4. with the aggregate results for **4 and 7 class** problem, averaged for 10 folds.

- Use a parametric unimodal (PU) method using the CMB distribution

We found your proposal very interesting and decided to implement the new parametric loss using a Poisson distribution based on your article: "Unimodal Probability Distributions for Deep Ordinal Classification". We named this loss as Poisson Unimodal (PU). We updated the manuscript by adding information about this loss in "Related Work" section and also by adding in all tables the results for this loss.

- Experiment with using a mixture distribution between p_u and p_s

We found your comment very interesting for future works, however we decided to not implement in this article because we thought we would deviate from the central focus of the article.

Other less significant points: - While it was appreciated that the authors tried out a vast range of architectures, perhaps it would make for a better presentation if the number of learnable parameters was stated for each of these architectures. You could then explore performance vs # parameters. It seems like the dataset you have used is extremely tiny, and having excessively large networks could degrade generalisation performance here. If it saves you computational resources, I don't think some of these architectures are strictly needed in the analysis: for instance AlexNet and VGG, which were superceded by ResNets (and for good reason). - It would be interesting to explore the case where you don't start off with pre-trained ImageNet weights. While I would expect such a network to very easily overfit, it can be controlled with sufficient regularisation (weight decay), and also allow you to explore the effect of having a severely constrained distribution (i.e. PU) in a 'low data' setting.

We agree with your concern, we also want to explore in future works the performance vs # parameters among the different architectures. We run our models across a large number of architectures to prove the robustness of our proposal loss regardless the architecture.

**Reviewer 3:**

**Experimental design** Only averages of 5 folds are given, in order to show the robustness I suggest to provide results of multiple experiments (e.g. 10), then averages and variance / standard deviation or box plots. And in addition I suggest to perform statistical significance tests on the prosed and tested algorithms.

**Validity of the findings** The proposed algorithm cost functions are straightforward. It would be a surprise if ordinal classification can benefit from these cost functions in general. Only a rigorous statistical evaluation of the proposed cost functions based on 10 or more data sets utilising more complex statistical evaluation (e.g. Wilcoxons test) in comparison with the other approaches could prove the strength of the proposed algorithm.

We agree with your concern and we implemented stratified 10-fold cross-validation and trained again all the models. We also updated the manuscript Tables of results. Furthermore, the results are compared to the best loss result and a statistical test is used with the hypothesis of them being the same or not. A $p$-value of 0.1 is used with a one-sided paired $t$-test.

---

## Round 0.3 · Minor Revisions

· Academic Editor

Minor Revisions

Your paper is basically ready for acceptance, but I have now also read through the paper myself (which I enjoyed!) and found a number of little wording and grammar problems that you should fix:

Abstract

"approaches at ordinal inference" -> "approaches to ordinal inference"

"one a multitude" -> "one of a multitude"

"not take advantage of the ordinal problem" -> "not sufficiently take advantage of the ordinal problem"

Introduction

"which justifies" -> ", which justifies"

Background

"This system rules" -> "This system assumes"

"can also include" -> "include"

"was the Herlev Dataset" -> "is the Herlev Dataset"

"in Herlev dataset" -> "in the Herlev dataset"

Related work

"on Herlev dataset:" -> "on the Herlev dataset." and remove paragraph break following this

"The Chankong’s approach" -> "The approach"

"the FCM" -> "the classifier" ???

"for the previous Chankong’s approach" -> "for the approach by Chankong et al."

"in the first step" -> "in the first step,"

"A different approach" -> "One approach"

"Binomial’s" -> "the Binomial"

"work is that" -> "is that"

"is exploring" -> "explore"

"also proposes the use" -> "also propose the use"

"mislabels" -> "mislabeled examples"

Proposal

"to a regularization term" -> "to add a regularization term"

"is controlling the relative importance" -> "controls the relative importance"

"equivalent to assume" -> "equivalent to assuming"

Experiments

"it was also evaluated" -> "we also evaluate"

"were implemented" -> "are implemented"

"were of different sizes" -> "are of different sizes"

"the dataset was augmented" -> "the training dataset was augmented" ???

"that successively reduces" -> "that successively reduce"

Delete "In the end, a series of outputs produce the desired classification."

Provide a reference for dropout.

"eight different" -- there are nine architectures in your list!!!

"9 different" -> "nine different"

"by these layers:" -> "by the following layers:"

"neuron outputs" -> "output neurons"

"The following ImageNet" -> "Subsequent ImageNet"

"which consisted mostly in an explodingly" -> "that consist mostly of a much"

"cost nearly doubling" -> "required nearly doubling"

How are the new layers you have added initialized?

Results

"starts with a" -> "started with a"

"Tables 3–8," -> "Tables 3–8, A1, and A2"

Delete "Appendix - table A1-A2".

".OE loss" -> ". OE loss"

"our loss, which" -> "our loss:"

"for 7-class problem and is tied for 4-class problem" -> "for the 7-class problem and is tied for the 4-class problem"

"with MAE metric" -> "with the MAE metric"

"Table A1 and Table A2 in appendix represent" -> "Tables A1 and A2 in the appendix present"

", being the ordinal losses in" -> ": the ordinal losses perform better"

"in appendix Table A3 and Table A4" -> "in Tables A3 and A4 in the appendix"

"In both Tables are represented" -> "Both tables present"

"Beckham and Pal (2017)" -> "(Beckham and Pal 2017)"

"ordinal-aware then" -> "ordinal-aware"

"could already be seen in Figure 6" -> "can also be seen in Figure 6"

"figure for eligibility" -> "figure for legibility"

"In average" -> "On average"

"using Herlev dataset" -> "using the Herlev dataset"

"the confusion matrix from Figure 5 contrasts" -> "the confusion matrices in Figure 5 contrast"

"in its characteristics" -> "in their characteristics"

"and ignoring all the other probabilities" -> "and ignore all the other probabilities"

"10 folds. (higher is better)." -> "10 folds (higher is better)."

"Non-parametric losses" -> "Comparing ordinal deep learning approaches on cervical cancer data, non-parametric losses"

"the existing in deep ordinal classification techniques when imposing uni-modality in probabilities distribution" -> "existing deep ordinal classification techniques that impose uni-modality in the probability distribution"

"the state-of-the-art by" -> "the state-of-the-art nominal-class approaches by"

"is pestered by two new hyperparameters, albeit" -> "requires two new hyperparameters. However,"

Delete "or even time-series problems"???

"accuracy performances" -> "accuracy"

"it is more complex" -> "data is more complex"

"so further work are" -> "so further work is"

"which allows during the train to" -> "which allows, during the training, to"

"for the access to Herlev Pap smear dataset" -> "access to the Herlev Pap smear dataset"

·

Basic reporting

no comment

Experimental design

no comment

Validity of the findings

no comment

Additional comments

Thanks for addressing my concerns. Good work.

---

## Round 0.4 · accepted · Accept

· Academic Editor

Accept

I do not have any further comments.